# *Autophagy-Related 2* Regulates Chlorophyll Degradation under Abiotic Stress Conditions in *Arabidopsis*

**DOI:** 10.3390/ijms21124515

**Published:** 2020-06-25

**Authors:** Zhuanzhuan Jiang, Li Zhu, Qiuyu Wang, Xin Hou

**Affiliations:** State Key Laboratory of Hybrid Rice, College of Life Sciences, Wuhan University, Wuhan 430072, China; zzjiang@whu.edu.cn (Z.J.); 2018202040068@whu.edu.cn (L.Z.); 2019202040063@whu.edu.cn (Q.W.)

**Keywords:** chloroplast degradation, autophagy, ATG2, high light intensity, high temperature, chloroplast vesiculation, chlorophyll degradation

## Abstract

Chloroplasts are extraordinary organelles for photosynthesis and nutrient storage in plants. During leaf senescence or under stress conditions, damaged chloroplasts are degraded and provide nutrients for developing organs. Autophagy is a high-throughput degradation pathway for intracellular material turnover in eukaryotes. Along with chloroplast degradation, chlorophyll, an important component of the photosynthetic machine, is also degraded. However, the chlorophyll degradation pathways under high light intensity and high temperature stress are not well known. Here, we identified and characterized a novel *Arabidopsis* mutant, *sl2* (*seedling lethal 2*), showing defective chloroplast development and accelerated chlorophyll degradation. Map-based cloning combined with high-throughput sequencing analysis revealed that a 118.6 kb deletion region was associated with the phenotype of the mutant. Complementary experiments confirmed that the loss of function of *ATG2* was responsible for accelerating chlorophyll degradation in *sl2* mutants. Furthermore, we analyzed chlorophyll degradation under abiotic stress conditions and found that both chloroplast vesiculation and autophagy take part in chlorophyll degradation under high light intensity and high temperature stress. These results enhanced our understanding of chlorophyll degradation under high light intensity and high temperature stress.

## 1. Introduction

Plants obtain energy for growth and development via photosynthesis and the oxidation of organic substrates by respiration [1]. During senescence onset or under stress conditions, photosynthesis is severely affected, and plant cells recycle damaged organelles to release nutrients to improve survivability [2,3]. Chloroplasts are extraordinary organelles in photosynthetic plants and contain 80% of the nitrogen in leaves. During leaf senescence or under stress conditions, damaged chloroplasts are degraded and provide nutrients for new leaves or developing organs [4]. In plants, chlorophylls, the most important natural pigments, combine with photosynthetic proteins to capture solar energy and transfer electrons. The dissociation of chlorophyll and its associated proteins is a necessary step for chloroplast degradation [5].

The chlorophyll cycle, including chlorophyll synthesis and chlorophyll degradation, is strictly regulated over the plant life period [6,7,8]. In *Arabidopsis*, the chlorophyll biosynthesis pathway contains 15 key enzymes encoded by 27 genes [9]. The chlorophyll degradation pathway involves the conversion of fluorescent molecules (chlorophyll a and chlorophyll b) to nonfluorescent chlorophyll catabolites (NCCs) via eight steps by ten genes [10]. Chlorophyll can also act as a potential signal to evoke reactive oxygen species (ROS) production [7,11]. In this case, there are five chlorophyll catabolism enzymes that convert fluorescent chlorophyll into safe chlorophyll catabolites [7]. The chlorophyll cycle is balanced with chloroplast development. During plant growth, chlorophyll content is balanced with chloroplast development and division, and the chlorophyll cycle is in a steady state. When the balance is broken by environmental or developmental cues, the chlorophyll content decreases, and chloroplast development is affected.

Recently, three pathways of chloroplast degradation were revealed: senescence-associated vacuoles (SAVs), chloroplast vesiculation, and autophagy [3,12,13,14]. SAVs are small protein lytic compartments containing chloroplast stroma-targeted fluorescent proteins that can be transported to vacuoles for degradation [15]. Chloroplast vesiculation, a new type of chloroplast degradation pathway, is independent of SAVs and autophagy and has been demonstrated to be involved in the degradation of chloroplast thylakoid-associated proteins [14,16]. There are three types of chloroplast component degradation via the autophagy pathway: the rubisco-containing body (RCB) [4], the Autophagy-related 8—interacting protein 1—green fluorescent protein labels plastid-associated body (ATI-PS body) [17], and small starch granule-like structures (SSGL) [18]. Rubisco is the most abundant stromal protein in chloroplasts, accounting for 12% to 30% of total leaf proteins in the C3 species. The degradation of chloroplast rubisco and other stromal proteins via small double-membrane autophagic vesicles has been connected to RCBs. Autophagy-related 5 (ATG5) and autophagy-related 7 (ATG7) participate in the production of RCBs [4,19]. The ATI-PS body is considered to be the other autophagic vesicle that contains chloroplast stromal, thylakoid, and envelope proteins. Autophagy-related 8 (ATG8) is involved in this pathway. Intriguingly, ATG8 participates in delivering the ATI-PS body to the vacuole but does not release the ATI-PS body from the plastid [17]. Chloroplasts are the main site of starch grain synthesis in plants. When autophagic activity is blocked via autophagy-related 4a/4b (ATG4a/4b) or autophagy-related 6 (ATG6) silencing, the number of vacuole-localized SSGLs declines remarkably. It has been suggested that autophagy also participates in the degradation of SSGLs [18]. Hence, it is especially essential to gain further insight into the interactions and relationships among the different degradation pathways to chloroplasts.

Autophagy is generally defined as a system highly present in organisms ranging from yeast (*Saccharomyces cerevisiae*) to mammalian and plant species. It is also a well-known catabolic process that nonselectively or selectively degrades dysfunctional or obsolete organelles and proteins [20,21]. Two main forms of autophagy have been described: microautophagy and macroautophagy. Microautophagy is characterized by the degradation of unnecessary cytoplasmic constituents by tonoplasts via direct vacuolar membrane invagination in eukaryotic cells (from yeast to plants). Macroautophagy is known as a process that degrades targeted cytoplasmic components or cargos by forming double-membraned vesicles called autophagosomes [20,22]. Genetic studies have identified more than 40 autophagy-related (ATG) genes required for various types of autophagy in yeast, among which 19 genes are involved in autophagosome formation [20,21,23]. These 19 ATG genes are classified into six functional groups: (i) the ATG1–ATG13 kinase complex, (ii) vesicles containing the transmembrane protein ATG9, (iii) ATG14 with the phosphatidylinositol 3-OH kinase (PI (3) K) complex, (iv) the ATG2–ATG18 complex, (v) the ATG12–ATG5 conjugation system, and (vi) the ATG8 lipidation system [21]. Previous studies have illuminated the mechanisms and physiological roles of the six functional groups in autophagosome formation in yeast.

However, unlike yeast, plant cells have more complicated structures and larger genomes. Genomic databases have identified 25 genes involved in autophagy in the *Arabidopsis* genome, corresponding to 12 genes in yeast [24], indicating that the members of the autophagy family expand in *Arabidopsis* compared with yeast. For example, ATG8 and ATG18 generally exist as a multiprotein family in *Arabidopsis.* The loss of function of ATGs in *Arabidopsis* may cause defective autophagy in *atg* mutants [3,25,26,27]. However, the functions of the redundant homologous genes in *Arabidopsis* are not yet clearly understood. Several ATG mutants including *atg2*, *atg5*, *atg18a* have been shown to exhibit an early senescence phenotype [3,25,26], and *atg2*, *atg7*, *atg9* and *atg18a* are more sensitive to abiotic stress, such as drought stress, salt stress, low oxygen, UV-B damage, oxidative stress, and nutrition starvation [24,28,29,30,31]. However, the relationship between autophagy defects and leaf color or chlorophyll degradation under abiotic stress conditions is largely unknown.

Here, we identified a seedling lethal mutant with albino cotyledon, abnormal chloroplast development and chlorophyll degradation. Map-based cloning combined with bulked segregant analysis revealed a 118.6 kb deletion containing 37 genes in the mutant. The loss of function of *ATG2* was responsible for accelerating chlorophyll degradation in *sl2* mutants, and described the abnormal chlorophyll cycle and disturbed photosystem of *atg2-1* mutants under growth conditions. Furthermore, we investigated *atg2-1* mutants under high light intensity and high temperature conditions to gain further insight into the relationships of chlorophyll and different chloroplast degradation pathways under stress conditions.

## 2. Results

### 2.1. Identification of the Seedling Lethal Mutant sl2

To investigate plant photosynthetic regulation and chloroplast development, we generated a collection of mutant lines of *Arabidopsis* (ecotype *Columbia-0*) and screened mutants with deficient chloroplast development phenotypes. Multiple seedling lethal mutants were obtained after extensive screening, and one of them had an albino cotyledon termed the *sl2* (*seedling lethal 2*) mutant. Because of the non-functional cotyledons, *sl2* could not survive without the supply of sucrose. When growing on ½ MS medium containing 1% sucrose, *sl2* mutants with albino cotyledon were obviously smaller than wild-type (WT) plants after one week of germination (Figure 1a). Four-week-old *sl2* mutant seedlings displayed a chlorophyll-deficient phenotype in true leaves (Figure 1a). Furthermore, compared with WT, *sl2* plants exhibited an aberrant photochemical capacity of photosystem II (F_v_/F_m_) at both the germination and seedling stages (Figure 1c, left). To further investigate their photosynthetic capacity, *sl2* mutants were grown on ½ MS medium containing 0%, 1%, 2%, or 4% sucrose. When grown without sucrose, the mutants could not undergo autotrophy and subsequently died. As the sucrose concentration increased (1%, 2%, and 4%), the mutants grew better even though they still had lower F_v_/F_m_ and higher nonphotochemical quenching (NPQ) than WT plants (Figure 1b), indicating that *sl2* mutants dissipated more excess excitation energy via the nonphotochemical pathway. Consistent with the photosynthetic defects, the increase in the fresh weight of *sl2* mutants was less than that of WT plants (Figure 1c, middle), and the degradation of chlorophylls had advanced in *sl2* mutants (Figure 1c, right).

Then, we observed the chloroplast ultrastructure of cotyledon and true leaves of WT plants and *sl2* mutants by transmission electron microscopy. The cotyledon chloroplasts of *sl2* mutants contained irregular and abnormal plastids with markedly reduced internal thylakoid membranes compared with those of WT plants (Figure 1d). The true leaf chloroplasts of *sl2* mutants could not divide normally, which resulted in enlarged chloroplasts. The chloroplast size of WT was 7.66 ± 0.77 µm, and the chloroplast size of *sl2* was 22.53 ± 7.3 µm (Figure 1e). Meanwhile, the accumulation of starch grains in *sl2* chloroplasts was significantly reduced compared with that in WT plants (Figure 1e), indicating that abnormal chloroplast division resulted in impaired function and limited the accumulation of starch grains.

Chlorophyll degradation could be accelerated by darkness [32,33]. After growing in the dark for seven days, *sl2* mutants seemed more etiolated than WT plants (Figure 1f), showing that dark treatment significantly enhanced the decline of chlorophyll content in *sl2* mutants. Chlorophyll combined with photosynthetic proteins constituted the major photosynthetic complexes. To investigate the possibility of whether chlorophyll degradation affects thylakoid membrane protein abundance, thylakoid membrane protein complexes from five-week-old WT and *sl2* seedlings were solubilized with n-dodecyl β-D-maltoside (DM) and analyzed by blue-native PAGE (BN-PAGE) to determine whether the thylakoid composition and/or assembly state of *sl2* mutants had changed. The assembly state of the thylakoid photosynthetic machinery of *sl2* seedlings was partial degradation (Figure 1g). Then, we performed immunoblotting analyses of the proteins in the thylakoid membrane complexes—PSII: D1, D2, PsbO, PsbQ and LHCII; photosystem I (PSI): PsaD; cytochrome *b_6_f* complex (Cyt*b_6_f*): Cyt*f*, *b_6_*, and PetC; and the CF_o_-CF_1_ complex: ATPα. The abundance of most of the PSII and cytochrome *b_6_f* complex proteins, including D1, D2, PsbO, PsbQ, LHCII, *b_6_*, and PetC, was decreased remarkably (Figure 1h). These results suggest that *sl2* mutants not only suffer from chloroplast dysplasia in cotyledons and true leaves but also from accelerated chlorophyll degradation and affected thylakoid proteins accumulation compared with WT plants, which lead to dramatic changes in the growth phenotype of *sl2* mutants.

### 2.2. Positional Cloning of SL2

To illustrate the genetic characteristics of the gene(s) controlling chloroplast development, the relevant gene(s) in *sl2* mutants were isolated by map-based cloning combined with bulked segregant analysis. A mapping population was generated by crossing the *sl2* mutants (with *Col-0* background) with *Ler* (*Landsberg erecta*). Using simple sequence length polymorphism (SSLP) markers [34] (Appendix A), *SL2* was located between the M10 and M11 markers on chromosome 3 (Figure 2a). Bulked segregant analysis was used for fine mapping [35]. The sequencing results showed that there was a 118.6 kb deletion in the region between the M10 and M11 markers in *sl2* mutants (Figure 2b). To confirm the deletion, three primers were designed (Figure 2c), and the PCR results showed that plants with albino cotyledon all had the deletion in the region (Figure 2d). The 118.6 kb deletion sequence contained 37 genes (Figure 2c, Appendix A) whose transcripts were not detected in the *sl2* mutants (Figure 2e). This indicates that the phenotype of the *sl2* mutant may be caused by the deletion of these genes.

Within the deletion region, 23 of the 37 annotated genes encoded unknown proteins, and the remaining 14 genes had been previously characterized (Appendix A). *At3g19220*, encoding *snowy cotyledon 2* (*SCO2*), regulates cotyledon chloroplast development, and its loss-of-function mutant shows an albino cotyledon phenotype. However, true leaves of *sco2* have a normal green appearance [36,37,38]. *AT3g19180*, encoding *the paralog of accumulation and replication of chloroplast 6* (*PACR6*), regulates chloroplast division, and its loss-of-function mutant shows abnormal chloroplast division [39,40]. *AT3g19190*, encoding *autophagy-related 2* (*ATG2*), regulates the early steps of autophagosome biogenesis, and its loss-of-function mutant shows autophagy defects [26,41]. Furthermore, we obtained Transfer DNA (T-DNA) insertion mutants (*sco2-1*, CS68145; *parc6-1*, SALK_100009C; and *atg2-1*, SALK_076727) (Figure 3a). The *sco2-1* mutants exhibited pale cotyledon but normal green true leaves (Figure 3b,c). Compared with the WT plants, the *parc6-1* mutants showed abnormalities in the size and shape of chloroplasts in mesophyll cells (Figure 3b). In particular, the phenotypes including pale cotyledon and larger chloroplasts in true leaves observed in *sl2* mutants might be due to the absence of SCO2 and PARC6, respectively (Figure 3b). In addition, the chlorophyll degradation in *sl2* mutants might result from ATG2 deficiency (Figure 3c). To investigate this possibility, we introduced homozygous *sl2* and *atg2-1* mutants with full-length *ATG2* cDNA driven by the cauliflower mosaic virus 35S promoter (*sl2*/35S::*ATG2* and *atg2-1*/35S::*ATG2*). T_2_ generation seedlings of four independent transgenic lines were analyzed. The degreening phenotypes of *sl2* and *atg2-1* were complemented by ATG2 in the transgenic plants (Figure 3d,e).

### 2.3. Lack of ATG2 Accelerated the Degradation of Chlorophyll and Thylakoid Proteins

*atg2-1* mutants provided ideal materials to investigate the relationship between autophagy and chlorophyll degradation. We measured the chlorophyll contents in the whole plant and functional leaves (greening leaves without chlorosis areas) of the WT and *atg2-1* mutant lines at different growth stages under favorable growth conditions, and found that the chlorophyll contents in both the whole plant and the functional leaves of three-week-old *atg2-1* mutants were similar to those of WT plants (Figure 4a). However, five-week-old *atg2-1* mutants had more degreening leaves, so their chlorophyll content dropped faster compared with WT plants (Figure 4a). We performed expression analyses at the transcriptional level, especially focusing on chlorophyll synthesis and chlorophyll degradation-related marker genes (Figure 4b,c). The qRT-PCR results showed that the transcription of all chlorophyll synthesis-related marker genes, including *Copper response defect 1* (*CRD1*), *divinyl chlorophyll vinyl reductase* (*DVR*), *Mg-protoporphyrin O-methyltransferase* (*CHLM*), *Mg-chelatase* (*CHLH*), *chlorophyll synthase* (*CHLG*), *pheophorbide a oxygenase (PORA)*, and *chlorophyllide a oxygenase* (*CAO*), were indistinguishable between five-week-old WT and *atg2-1* seedlings both in total plants and functional leaves, indicating that the chlorophyll synthesis-related marker genes were transcribed at the same level (Figure 4b). In contrast, the transcript levels of the chlorophyll degradation-related marker genes *pheophytinase* (*PPH*) and *Non-yellowing* (*NYE*) increased eight times and 16 times, respectively, in five-week-old *atg2-1* mutant plants compared with WT plants. The transcription of chlorophyll degradation-related marker genes *Non-yellow coloring 1* (*NYC1*)*, NYC1-like* (*NOL*)*, Accelerated cell death 1* (*ACD1*)*, Accelerated cell death 2* (*ACD2*) and *7-hydroxymethyl chlorophyll a reductase* (*HCAR*) more than doubled in *atg2-1* mutants compared with WT in whole plants (Figure 4c, left). However, the transcription of chlorophyll degradation-related marker genes was not significantly changed in the functional leaves of *atg2-1* mutants compared with those of WT plants, although the transcript levels of *PPH* and *NYE* showed a small change (Figure 4c, right). Taken together, these results suggested that the chlorophyll degradation signal was activated such that it accelerated chlorophyll degradation in the *atg2-1* mutants. Nevertheless, the assembly state of the thylakoid photosynthetic machinery of WT and *atg2-1* seedlings was not significantly different (Figure 4d). The abundance of the PSII proteins, especially D1 and D2, was decreased (Figure 4e). This result indicates that ATG2 deficiency accelerates not only the degradation of chlorophyll but also the abundance of PSII proteins.

### 2.4. ATG2 Was Expressed Ubiquitously and Induced by Abiotic Stresses

The analysis above showed that *atg2-1* mutants exhibited a degreening phenotype under normal growth conditions. Developmental processes as well as various stresses are the internal and external factors leading to leaf senescence. To explore the function of ATG2 under abiotic stresses, we used Plant CARE software [42] to analyze the promoter region 2000 bp upstream of the ATG2 start codon. As shown in Figure 5a, the existence of cis-acting regulatory elements on an abscisic acid responsive element, a MeJA-responsive element, a light responsive element, and a drought-responsive element implied that *ATG2* might be regulated by abscisic acid (ABA), jasmonic acid (JA), light, drought or other abiotic stresses. Then, we examined the expression pattern of *ATG2* by using quantitative RT-PCR (qPCR) analysis. The results show that *ATG2* was detected in all tissues, and the expression level was slightly higher in flowers than in other tissues (Figure 5b). To investigate whether ATG2 responded to abiotic stress, we studied the expression level of *ATG2* under the abiotic stresses of high light intensity (350 μmol·m^−2^·s^−1^) and high temperature (30 °C). *ATG2* had a fluctuating expression pattern in WT plants, and there was one high expression peak when the plants were transferred to high light intensity conditions for 24 h. Moreover, similar results were observed when the plants were transferred to high temperature conditions, but there was no significant expression peak. Therefore, these results reveal that *ATG2* was induced by the abiotic stresses of high light intensity and high temperature.

### 2.5. Atg2-1 Was Highly Sensitive to High Light Intensity

The expression patterns of *ATG2* in response to different abiotic stresses suggested that ATG2 was involved in the physiological responses to these stresses. To test this hypothesis, *atg2-1* mutants were exposed to different abiotic stresses. No obvious difference was observed between WT and *atg2-1* seedlings when they were grown on ½ MS medium for ten days under normal light (80 μmol·m^−2^·s^−1^). After continuous exposure to high light intensity conditions for three days, *atg2-1* mutants exhibited a necrosis phenotype in the cotyledons, but the difference from WT plants was not significant. However, when exposed to high light intensity (350 μmol·m^−2^·s^−1^) conditions for six days, *atg2-1* mutants showed a necrotic phenotype in older leaves (1st and 2nd cycle of rosettes) (Figure 6a). In accordance with the visible phenotype, the leaves of *atg2-1* mutants showed significantly lower chlorophyll retention, fresh weight, and F_v_/F_m_ ratio than WT plants when exposed to high light intensity conditions for six days (Figure 6b). To examine the relationship between lower chlorophyll retention and chlorophyll breakdown in *atg2-1* mutants after high light intensity treatment, we performed expression analyses at the transcriptional level, especially focusing on chlorophyll degradation-related marker genes. However, WT plants and *atg2-1* retained the same transcription of chlorophyll degradation-related marker genes under high light intensity conditions (Figure 6c). Intriguingly, when grown under growth light intensity (80 μmol·m^−2^·s^−1^), the phenotype of *atg2-1* mutants was indistinguishable from that of WT plants after three weeks of germination. However, after exposure to high light intensity (350 μmol·m^−2^·s^−1^), the rosette leaves of *atg2-1* mutants accumulated more anthocyanin, which was upregulated in the presence of biotic and abiotic stresses [43], than those of WT seedlings (Figure 6d). In quantitative terms, the rosette leaves of WT plants accumulated 100 μg/g of total anthocyanin, whereas *atg2-1* mutants accumulated 150 μg/g, an increase of almost 50% (Figure 6e). In plants, reactive oxygen species (ROS) are signal transduction molecules that induce a series of downstream responses, such as chlorophyll degradation and anthocyanin biosynthesis [11,44,45]. Therefore, we further examined the expression levels of genes responsive to ROS, including *Toll-interleukin resistance* (*TIR*), *Trypsin inhibitor protein 1* (*TIP1*), *Stress-associated protein 12* (*SAP12*) and *Tol B protein-like protein* (*TPL*). Before the treatment, the expression levels of the four selected marker genes were similar in the WT plants and *atg2-1* mutants. When grown under high light intensity conditions for three days or six days, the expression levels of these genes were significantly higher in *atg2-1* mutants than in WT plants (Figure 6f). Thus, high light intensity might cause more extensive oxidative damage in *atg2-1* mutants than WT plants due to autophagy defects.

Previous studies indicated that the formation, transportation, and fusion of complete autophagosomes involved the coordination of a set of autophagic family members [21]. Likewise, chloroplast vesiculation has been demonstrated to degrade chloroplast thylakoid-associated proteins under stress conditions [14,46]. However, the degradation pathways associated with *chloroplast vesiculation* remain unclear, with *chloroplast vesiculation* just a singular defined genetic entity in our research. There is insufficient evidence that SAV is completely independent from the autophagy pathway [15]. We examined the mRNA levels of five autophagic family genes (*ATG6*, *ATG18A*, *ATG8A*, *ATG5*, and *ATG12A*) and the *chloroplast vesiculation* gene under high light intensity conditions. In WT plants, the mRNA levels of *ATG8A*, and *ATG12A* were not significantly different, but the mRNA levels of *ATG6*, *ATG5* and *ATG18A* were increased four times under high light intensity conditions (Figure 6g). While the autophagy pathway was blocked in *atg2-1* mutants, the mRNA levels of *ATG6*, *ATG5*, *ATG12A* and *ATG18A* were increased, too. The mRNA level of *chloroplast vesiculation* was increased eight times in WT plants after transfer to high light intensity conditions (Figure 6g). While the autophagy pathway was blocked in *atg2-1* mutants, the mRNA level of the *chloroplast vesiculation* gene was more highly induced than in the WT plants (approximately 6 times) after transfer to high light intensity conditions (Figure 6g). The results suggested that the autophagy and *chloroplast vesiculation* pathways may play a role under high light intensity conditions for chloroplast degradation.

### 2.6. atg2-1 Enhanced High Temperature Induced Chlorophyll Degradation

The effects of ATG2 on tolerance to high temperature were examined by comparing the growth of *atg2-1* mutants with that of WT plants grown in ½ MS plates and soil at 30 °C. As shown in Figure 7a, *atg2-1* mutants were not significantly different from WT plants under growth temperature (22 °C). However, the fresh weight of mutants was significantly lower than that of WT plants when grown at 30 °C for seven days and for 14 days, and the fresh weight and chlorophyll retention of *atg2-1* mutants was dramatically decreased compared with those of WT plants (Figure 7b). However, the photochemical capacity of PSII (F_v_/F_m_) declined at similar rates under high temperature stress in WT seedlings and *atg2-1* mutants. Likewise, the transcription of chlorophyll degradation-related marker genes was detected. As shown in Figure 7c, an increasing tendency in the transcript level was measured for several target genes (such *PPH*, *ACD2*, *NYE* and *HCAR*) that responded to the high temperature treatments. The same phenotype was displayed in *atg2-1* mutants when grown in soil. Upon exposure to high temperature stress (30 °C) for seven days, the older leaves of *atg2-1* mutants yellowed, but the older leaves of WT plants did not (Figure 7d). In quantitative terms, the chlorophyll retention and fresh weight of *atg2-1* mutants both decreased to one third of those of the WT plants (Figure 7e). Taken together, these results suggested that *ATG2* deficiency made the plant more etiolated to high temperature. Noticeably, when exposed to high temperature conditions, the mRNA levels of *ATG6*, *ATG18A*, *ATG5*, and *ATG12A* were increased significantly in WT plants. When the autophagy pathway was blocked in *atg2-1* mutants, the mRNA level of some members of the autophagy pathway (*ATG6*, *ATG5*, *ATG12A*) was increased (Figure 7f). The mRNA level of the *chloroplast vesiculation* gene was increased nine times in WT plants after transfer to high temperature conditions (Figure 7f). While the autophagy pathway was blocked in *atg2-1* mutants, the mRNA level of *chloroplast vesiculation* was more highly induced than WT plants (approximately 2 times) after transfer to high temperature conditions (Figure 7f). The results suggested that the autophagy and *chloroplast vesiculation* pathways may play a role in chlorophyll degradation under high temperature conditions.

### 2.7. ATG2 Interacts with ATG18A in Arabidopsis

Based on phylogenetic analysis, AtATG2 was the only homolog of yeast ATG2p. In yeast, ATG2p and ATG18p interact during autophagosome formation [21]. In *Arabidopsis*, there are eight ATG18p homologs named ATG18A–ATG18H. ATG18A–ATG18H have one to three WD-40 repeat proteins and are expressed in various organs and tissues of plants [25]. ATG18B showed the highest similarity with ATG18p and ATG18G showed the lowest similarity based on the sequence comparisons of the full-length proteins (Figure 8a). However, there were no obvious phenotypes of *atg18b* mutants, during nitrogen and carbon starvation and during senescence. *atg18a* mutants showed early leaf senescence and were more sensitive to nutrient starvation conditions [25]. We further examined the interaction between ATG2 and ATG18A–ATG18H using yeast two-hybrid assays. ATG2 was fused with the GAL4 transcription activation domain (AD), and ATG18A–ATG18H were fused with the GAL4 DNA-binding domain (BD) and transformed into AH109. The results showed that ATG2 could interact with ATG18A but not with other ATGs in yeast. To confirm the interaction, ATG2 was fused with BD and ATG18A with AD. The results indicated that AtATG2 interacted with ATG18A in yeast (Figure 8b).

## 3. Discussion

In green plants, chloroplasts are not only photosynthetic organelles but also store a pool of nutrients, particularly nitrogen. Thus, the degradation of intracellular components damaged by high light intensity or other environmental factors is a crucial approach to protect photosynthetic machinery and nitrogen recycling for growing organs [3]. Three chloroplast degradation pathways have been demonstrated so far: autophagy, SAVs, and chloroplast vesiculation. Autophagy plays critical roles in the bulk degradation of cytoplasmic components and organelles throughout the plant life cycle, but SAVs and chloroplast vesiculation are specialized degradation pathways for chloroplasts [47]. SAVs are responsible for degrading stromal proteins in plastids, excluding rubisco. Chloroplast vesiculation is responsible for degrading thylakoid proteins involved in the photochemical apparatus. Chlorophyll breakdown via five chlorophyll catabolic enzymes is required for chloroplast degradation [7,48]. ATG2 is a member of the autophagy family, known as a major system for clearing both surplus and dysfunctional organelles, and the mechanisms have been well elucidated in yeast and animals [49]. As reported, mutants that completely lack autophagy show early senescence [3,25]. In this study, the autophagy-defective mutant *atg2-1* accelerated the degradation of chlorophyll along with an increasing growth period under normal conditions (Figure 4a). The mRNA level of chlorophyll degradation-related marker genes markedly increased but the chlorophyll synthesis-related marker genes remained unchanged (Figure 4b,c). Although the photosystem complex assembly was not impacted (Figure 4d), the abundance of PSII proteins (such as D1 and D2) was reduced (Figure 4e). Our results suggested that chlorophyll degradation was enhanced when plant autophagy was absent. A variety of stress responsiveness elements were distributed upstream of the *ATG2* gene translation start codon (Figure 5a), and the expression level of *ATG2* could be induced by stress conditions. These results indicated that ATG2 play a crucial role under high light intensity and high temperature stress conditions.

Due to the characteristics of sessile organisms, plants have evolved complex but orderly degradation mechanisms to confront different levels of environmental stresses. Light and temperature are environmental factors necessary for the growth of plants, and when light and temperature reach extremes, they can cause damage to plants. Chloroplasts are directly driven by light in a process called photorelocation to capture more energy under low light intensity and to avoid photodamage under high light intensity [50]. High light intensity can cause damage to photosynthetic proteins, especially the PSII reaction center protein D1, leading to a reduction in photosynthesis efficiency. When the D1 protein is damaged, a newly synthesized D1 protein replaces the damaged D1 protein. This turnover regularly occurs in chloroplasts but occurs more frequently under high light intensity conditions [51]. However, temperature involves numerous biochemical reactions, metabolite synthesis, and signal transduction, and temperature can also affect the fluidity of lipids, which is the necessary basis of the structure of cell membranes and of various reactions in cells [1]. For chloroplasts, high temperature increases the fluidity of thylakoid membranes, resulting in the removal of PSII light-harvesting complexes from the thylakoid membrane [52]. Meanwhile, high temperature affects the stability of light-harvesting chlorophyll-protein complexes and the deactivation of photosynthetic enzymes [53]. Therefore, high light intensity and high temperature cause different levels of stress to the photosynthetic apparatus [54]. Recent studies have indicated that *atg* mutants turned yellow after hypoxia treatment [28], and exhibit photobleaching in rosette leaves when exposed to UV-B damage [29]. A large portion of the dysfunctional chloroplasts are transported into the vacuole via the entire organelle type, called chlorophagy, while a small portion of the dysfunctional chloroplasts are transported via RCB [29]. However, other degradation pathways that are independent of autophagy may be involved in the chlorosis phenotype. In our study, when exposed to high light intensity, the chlorophyll content of *atg2-1* mutants decreased (Figure 6a), but the relative expression levels of *chloroplast vesiculation* and autophagic genes increased markedly after treatment for one to two days and then decreased rapidly (Figure 6g). High light intensity treatment also led to anthocyanin accumulation (Figure 6d,e), and the expression levels of ROS-related marker genes were higher in mutant plants (Figure 6f). However, the expression levels of chlorophyll degradation-related marker genes in mutants were similar to those in WT plants (Figure 6c). The expression of autophagy-related genes was induced under high light intensity (Figure 6g), it was consistent with previous research [29] that autophagy plays a role in high light intensity. Our data show that the expression of *chloroplast vesiculation* was highly induced by high light intensity (Figure 6g), suggesting that chloroplast vesiculation is also responsible for high light intensity-induced chlorophyll degradation. When exposed to high light intensity, *chloroplast vesiculation* was much more highly induced in *atg2-1* mutants than in WT plants (Figure 6g). This result indicates that when autophagy cannot work, plants need more chloroplast vesiculation to work on chlorophyll degradation.

In addition, when exposed to high temperature, the chlorophyll content of *atg2-1* mutants was decreased (Figure 7a). Correlated with the high-temperature hypersensitive phenotype, the relative expression level of chlorophyll degradation-related marker genes increased much more highly in the mutants than in WT (Figure 7c). Similar to high light intensity treatment, high temperature also induced the expression of autophagy-related genes and the *chloroplast vesiculation* gene, and *chloroplast vesiculation* was more highly induced in *atg2-1* mutants than WT plants (Figure 7f). It could be concluded that plants might degrade chloroplasts by autophagy and chloroplast vesiculation under both high light intensity and high temperature conditions, and that plants could select optimal approaches to remove the useless molecules in response to different abiotic stress environmental or developmental circumstances.

Previous studies have revealed that ATG2p and ATG18p localize to the pre-autophagosomal structure (PAS), forming complex preautophagosomal membranes similar to the autophagosomes in yeast. PAS has been identified as a functional entity responsible for autophagosome formation [21,55]. Molecular genetics studies have revealed homologs of yeast autophagy genes in *Arabidopsis*. Interestingly, some yeast autophagy genes were found not to have homologs in *Arabidopsis* (such as *ATG11*, *ATG17*, and *ATG19*), while some yeast autophagy genes have multigene families in *Arabidopsis* (such as *ATG8*, *ATG18*, and *ATG21*) [56]. There is a single homolog of yeast ATG2p in *Arabidopsis* and eight homologs of yeast ATG18p [25,57]. According to sequence similarity, ATG18B, not ATG18A, in *Arabidopsis* is more similar to ATG18p in yeast. *Atg18a* mutants showed an early-senescence phenotype and were more sensitive to nutrient deprivation conditions, similar to ATG2 mutants [25]. Our data reveal that ATG2 only interacts with ATG18A (Figure 8b). This result indicates that the biological function of autophagy genes in *Arabidopsis* might not completely correspond to the sequence similarity of yeast autophagy genes.

A *sl2* mutant has been identified that has a 118.6 kb deletion region containing 37 genes. Thirteen of the genes were suggested to occupy an important position in plants, regardless of structural development or regulatory function (Appendix A). Although the mutants exhibited growth defects, abnormal chloroplast development and early senescence, they could still survive and grow. Furthermore, our results provide evidence that autophagy plays a significant role in maintaining the balance of different chlorophyll degradation pathways under normal or stress conditions. This raises the following question: how do plants distribute survival pressure? There is no doubt that plants have evolved various pathways to mitigate risks with long-term evolutionary selective pressure. Increasing evidence indicates that genes are regularly distributed on chromosomes in eukaryotes [58,59]. Moreover, for vital organelles, such as chloroplasts, plants have developed complex and sophisticated degradation systems to minimize damage and maximize survival. Thus, it may be an interesting project to explore survival strategies in plants.

## 4. Materials and Methods

### 4.1. Plant Materials and Growth Conditions

*Arabidopsis* ecotype *Columbia-0* (*Col-0*) and *Landsberg erecta* (*Ler*) plants were used in the study. The T-DNA insertion mutant lines (SALK_100009C, locus AT3g19180; SALK_076272, locus At3g19190; CS68145, locus AT3g19220) were obtained from the *Arabidopsis* Resource Center (Columbus, OH, USA). For soil-grown plants, sown seeds were cold-treated for two days and then transferred to the indicated growth conditions (22 °C, 16 h light/8 h dark, 80 μmol·m^−2^·s^−1^). For plate-grown plants, surface-sterilized seeds were sown on ½ MS (half-concentration of normal Murashige and Skoog) medium with 0.8% agar, cold-treated for two days, and grown under the same conditions as soil-grown plants.

### 4.2. Transmission Electron Microscopy

Seven-day-old cotyledons and three-week-old true leaves were collected and fixed in 2.5% glutaraldehyde according to standard procedures as outlined in [60]. The slot copper grids were observed under a transmission electron microscope (JEM-1400plus, JEOL, Tokyo, Japan).

### 4.3. Measurement of Chlorophyll Fluorescence and Chlorophyll Content

Chlorophyll fluorescence parameters were measured by a Fluor Cam800-C (Photon System Instruments, Prague, CZ) after 30 min dark adaption. The F_v_/F_m_ ratio was defined as (F_m_ − F_o_)/F_m_, and the nonphotochemical quenching (NPQ) was calculated as (F_m_ − F’_m_)/F’_m_, where F_m_ is the maximum fluorescence value in the dark-adapted state; F’_m_ is the maximum fluorescence value in any light-adapted state; and F_o_ is the minimal fluorescence value in the dark-adapted state. The number of plants used in this experiment is indicated in the figure legend.

For pigment extraction, plant tissues (approximately 0.2 g of fresh weight) were pulverized in liquid nitrogen and extracted with 80% acetone. After centrifugation at 16,000 g for 5 min, the liquid phases were collected in a two-mL test tube. Spectrophotometric quantification was carried out in a NanoDrop 2000 (Thermo Fisher, Massachusetts, USA) using the following calculations: Chla = 12.21 × A_663_ − 2.81 × A_645_, and Chlb = 20.13 × A_645_ − 5.03 × A_663_ (μg/mL).

### 4.4. Measurement of Anthocyanin

Plant tissues (the number of plants used in this experiment is indicated in the figure legend) were pulverized in liquid nitrogen and dissolved in HCl (pH 2.0). After centrifugation at 16,000 g for 5 min, the supernatant was diluted with 0.4 M sodium acetate (pH 4.5) and 25 mM potassium chloride (pH 1.0). Total monomeric anthocyanin was measured by recording A_520_ as described by Giusti and Wrolstad [61].

### 4.5. Isolation and Mapping of the SL2 Locus

The large fragment deletion locus was identified by map base cloning combined with bulked segregant analysis (BSA). The *sl2* mutants (with *Col-0* background) were crossed with *Ler* (with *Landsberg erecta* background) seedlings to generate the F_1_ population. A total of 60 mutant individuals selected from the F_2_ populations were used for primary mapping. BSA was used for fine mapping, the *sl2* mutants were crossed with *Ler* and *Col-0* plants, and the resulting F_1_ hybrids were self-crossed to produce F_2_ seeds. The F_2_ populations were divided into two groups according to phenotypes: albino cotyledon and green cotyledon. Three F_2_ populations were selected as three method pools: *sl2* mutants crossed with *Ler* plants with albino cotyledon and *sl2* mutants crossed with *Col-0* plants with albino or green cotyledon. For each F_2_ population, DNA extracted from 40 individual plants was used to construct the library using a DNA Library Prep Kit (Thermo Fisher, Waltham, MA, USA). High-throughput sequencing was performed by an Illumina Hiseq 4000. After obtaining the original sequence, the joint sequence, PolyA, PolyN, and other sequences were filtered to get clean data. Finally, the clean data were analyzed to excavate the mutation site. The large fragment deletion test was PCR-amplified by using the primers F, R1 and R2. The primers are shown in Appendix A.

### 4.6. RNA Extraction, Reverse Transcription and Quantitative RT-PCR Assays

Total RNA was isolated using TRIzol Reagent (Invitrogen, California, USA) according to the manufacturer’s instructions. Reverse transcription was performed with the Prime Script™ RT Reagent Kit (TaKaRa, Tokyo, Japan). Quantitative real-time polymerase chain reaction (quantitative RT-PCR) assays were performed on a 7300 plus Real-Time PCR system (Applied Biosystems, Foster City, CA, USA) with SYBR Premix Ex Taq (TaKaRa, Tokyo, Japan). The *Actin2* gene was used as the endogenous control. The relative expression was calculated by using the formula 2^−ΔΔ*C*t^. All experiments were performed for two biological replicate and three technical replicates (primers used for quantitative RT-PCR are listed in Appendix A).

### 4.7. Chloroplast Thylakoid Proteins Isolation

Rosette leaves of five-week-old plants were briefly homogenized in grinding buffer (50 mM HEPES-KOH, 330 mM sorbitol, 10 mM EDTA, and 0.05% BSA, pH 8.0), and filtered with nylon mesh. Crude plastids were then collected by centrifugation at 7500× *g* for 10 min 4 °C. After centrifugation, chloroplast pellets were scoured gently by Sorbitol and HEPES buffer (50 mM HEPES-KOH, 330 mM sorbitol, pH 8.0), flowed by centrifugation at 7500× *g* for 10 min at 4 °C. Chloroplasts were subsequently lysed in lysis buffer (10 mM HEPES-KOH, pH 8.0) for 10 min, and then centrifuged at 7500× *g* at 4 °C, for 10 min. Finally, chloroplast thylakoid proteins were diluted to 1 mg/mL by adding SH buffer.

### 4.8. Blue-Native PAGE and the Second Dimension SDS-PAGE

Blue-native PAGE was performed as described in Kügler et al. [62]. Chloroplast thylakoid proteins were resuspended in 25BTH40G (25 mM BisTris-HCl, 20% (*w*/*v*) glycerol, pH 7.0) to a final chlorophyll concentration of 0.5 mg/mL. Thylakoid membranes were solubilized with solubilization buffer (25 mM BisTris-HCl, 20% (*w*/*v*) glycerol, pH 7.0, 2% n-dodecyl β-d-maltoside) on ice for 25 min, and unsolubilized material was removed by centrifugation at 12,000× g at 4 °C for 15 min. BN-sample buffer (100 mM BisTris-HCl, pH 7.0, 0.5 M 6-amino-caproic acid, 30% sucrose, 50 mg/mL Coomassie Brilliant Blue G 250) was added to the soluble protein. Proteins were loaded on native gradient page gels (4% to 16% acrylamide) for electrophoresis for 6–8 h. For 2D-SDS PAGE, individual lanes from the BN-PAGE gel were sliced cautiously with a blade and submerged in 2× SDS loading buffer (100 mM/L Tris-HCl, pH 6.8, 4% SDS, 0.2% bromophenol blue, 20% (*w*/*v*) glycerol, 20% β-mercaptoethanol) for 15 min at 75 ℃. The gel was embedded on top of 12% SDS-PAGE for electrophoresis. After electrophoresis, the proteins were visualized by Coomassie Brilliant Blue R 250 staining.

### 4.9. Immunoblot Analyses

Chloroplast thylakoid proteins were denatured at 95 °C for 15 min and then were separated by 12% SDS-PAGE and transferred to nitrocellulose membranes. Membranes were blocked with 5% nonfat dried milk. After blocking, the membranes were subsequently incubated with antibodies generated against the indicated proteins (D1, D2, PsbO, PsbP, PsbQ, LHCII, PsaA, PsaD, Cyt*f*, *b_6_*, PetC, and ATPα). All antibodies were generated in rabbits with polypeptides as the antigens. The horseradish peroxidase activity of rabbits’ secondary antibodies was detected using an electrochemiluminescence kit (Beyotime, Shanghai, China) according to the manufacturer’s instructions.

### 4.10. Complementation of the sl2 and atg2-1

The *AtATG2* full-length cDNA was amplified from *Arabidopsis* cDNA using a primer pair (Appendix A). The resulting fragment was ligated into the transient expression vector pCambia1300-35, which is under the transcriptional control of the cauliflower mosaic virus 35S promoter and has omega sequences for efficient translation. The plasmids were transferred into *Agrobacterium tumefaciens* (strain GV3101) by the freeze–thaw method. The floral buds of *sl2* and *atg2-1* mutant plants were transformed according to [63]. The transformants were selected on ½ MS medium containing 50 mg/mL hygromycin B.

### 4.11. Yeast Two-Hybrid Analysis

Yeast (*Saccharomyces cerevisiae*) two-hybrid assays were performed according to the Matchmaker Yeast Two-Hybrid System manual (Clontech, Tokyo, Japan). The full-length *ATG2* and *ATG18* family genes were cloned into pGADT7 (GAL4 activation domain) and pGBKT7 (GAL4 DNA-binding domain), respectively. The yeast strain AH109 was cotransformed with the constructs for each species using the polyethylene glycol/lithium acetate method and grown on the synthetic dropout medium SD-trp-leu for selection. Details about the primers used for plasmid construction are shown in Appendix A.

## Figures and Tables

**Figure 1 ijms-21-04515-f001:**
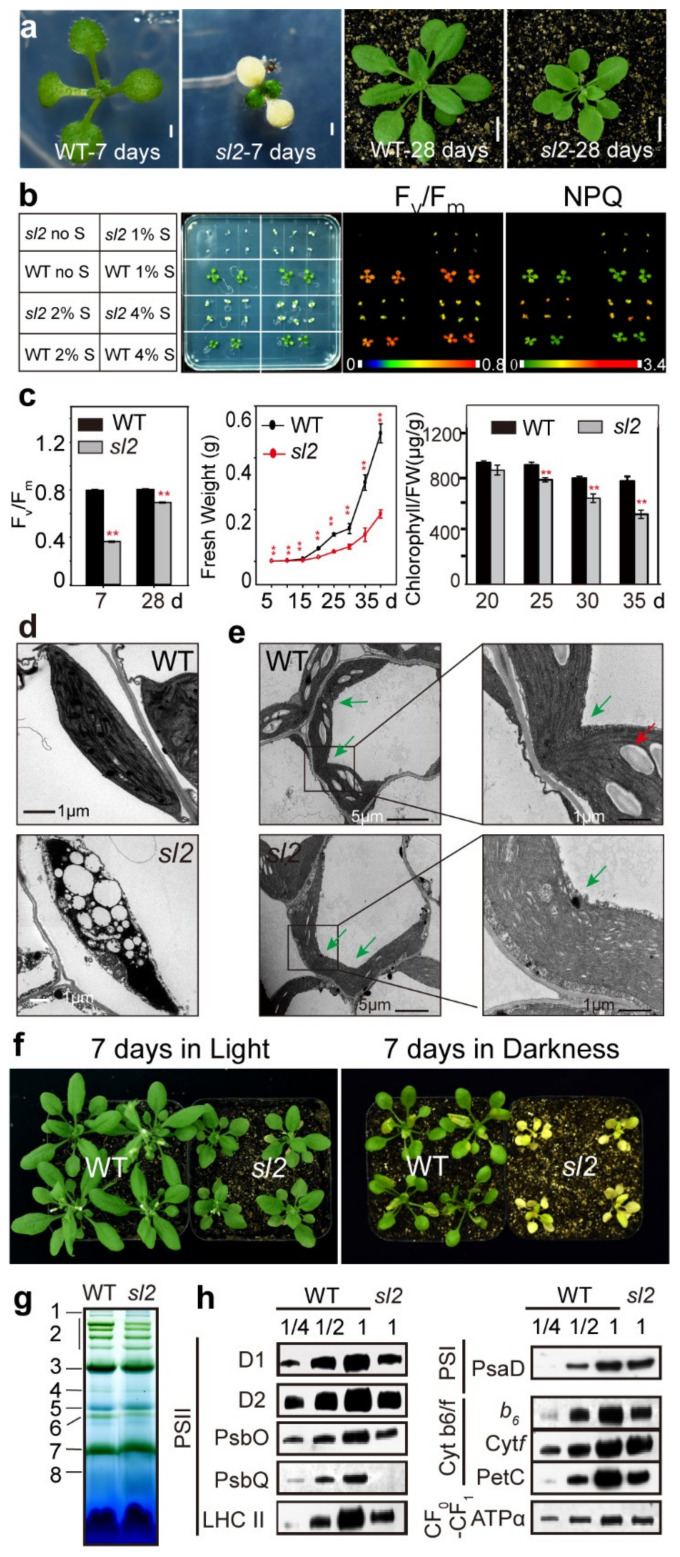
Phenotypic characterization of the *sl2* mutant. (**a**) Seedlings of the wild-type (WT) and *sl2* grown on ½ MS medium containing 1% sucrose for seven days (left, bar = 1 mm) and in soil for 28 days (right, bar = 1 cm); (**b**) Chlorophyll fluorescence images of the WT and *sl2* seedlings grown on ½ MS medium without or with 1%, 2%, and 4% sucrose. The false-color images ranging from black (0) to red (1) represent F_v_/F_m_, and green (0) to red (3.4) represent nonphotochemical quenching (NPQ); (**c**) The chlorophyll fluorescence (left), fresh weight (middle), and chlorophyll contents (right) of WT and *sl2* plants grown in soil. F_v_/F_m_, the maximum efficiency of PSII photochemistry. Data shown as mean ± SE (*n* = 8 for chlorophyll fluorescence, *n* = 12 for fresh weight, and *n* = 8 for chlorophyll content). Three independent experiments were performed with three biological replicates. The three biological replicates showed similar results, and one of them is shown in this figure. Asterisks show significant difference from the WT plants at ** *p* < 0.01 (Student’s *t* test); (**d**) Transmission electron micrograph of chloroplasts from cotyledons of seven-day-old WT and *sl2* seedlings grown on ½ MS medium containing 1% sucrose; (**e**) Transmission electron micrograph of chloroplasts from true leaves of three-week-old WT and *sl2* seedlings. Red arrows indicate starch grains; green arrows indicate chloroplast division sites; (**f**) WT and *sl2* seedlings grown in soil under normal growth conditions (16 h light/8 h dark, 80 μmol·m^−2^·s^−1^) for three weeks, and then cultured under normal growth conditions (left), or in darkness (right) for another week; (**g**) Blue native polyacrylamide gel analysis of five-week-old WT and *sl2* mutant plants (20 wild-type plants and 40 *sl2* mutants were used in this experiment). Annotation of the different complexes: 1, NDH-PSI; 2, PSII super complexes; 3, PSI monomer, PSII dimer, and PSII monomer with LHCII trimer; 4, PSI monomer and CF_1_ complex; 5, PSII monomer; 6, LHCII assembly; 7, LHCII trimer; 8, LHCII monomer; (**h**) Analysis of thylakoid membrane protein accumulation in the five-week-old WT and *sl2* mutant plants (about 20 wild-type plants and 40 *sl2* mutants were involved in this experiment). 4 μg chlorophyll of thylakoid proteins were separated by 12% SDS PAGE and immunoblots were performed using antibodies as indicated.

**Figure 2 ijms-21-04515-f002:**
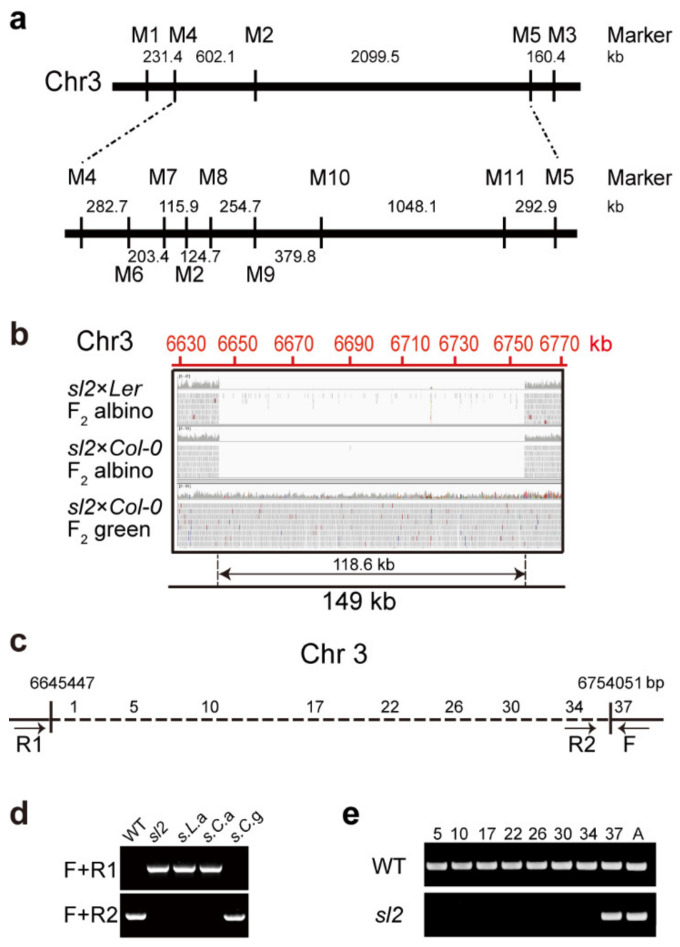
Isolation and mapping for albino cotyledon locus. (**a**) Schematic representing the map-based cloning of *sl2* mutants; M1-M11 are molecular markers for map-based cloning; (**b**) Bulked segregant analysis of *sl2* mutants; (**c**) Schematic diagram of genes in 118.6 kb deletion region; arrows indicate the primers used in (**d**). 1, 5, 10, 17, 22, 26, 30, 34, and 37 are the serial numbers of genes in the deletion region; (**d**) PCR analysis of genomic DNA from the WT and *sl2* mutants showed a 118.6 kb deletion in the mutant. Annotation of the different samples: s.L.a, *sl2* × *Ler* F_2_ albino seedlings; s.C.a, *sl2* × *Col-0* F_2_ albino seedlings; s.C.g, *sl2* × *Col-0* F_2_ green seedlings; (**e**) The expression analysis of the genes (5-*AT3G19220*, 10-*AT3G19270*, 17-*AT3G19300*, 22-*AT3G19340*, 26-*AT3G19380*, 30-*AT3G19420*, 34-*AT3G19460*, 37-*AT3G19490*) by RT-PCR. A, *actin* 2. All experiments were performed for two biological replicates and three technical replicates.

**Figure 3 ijms-21-04515-f003:**
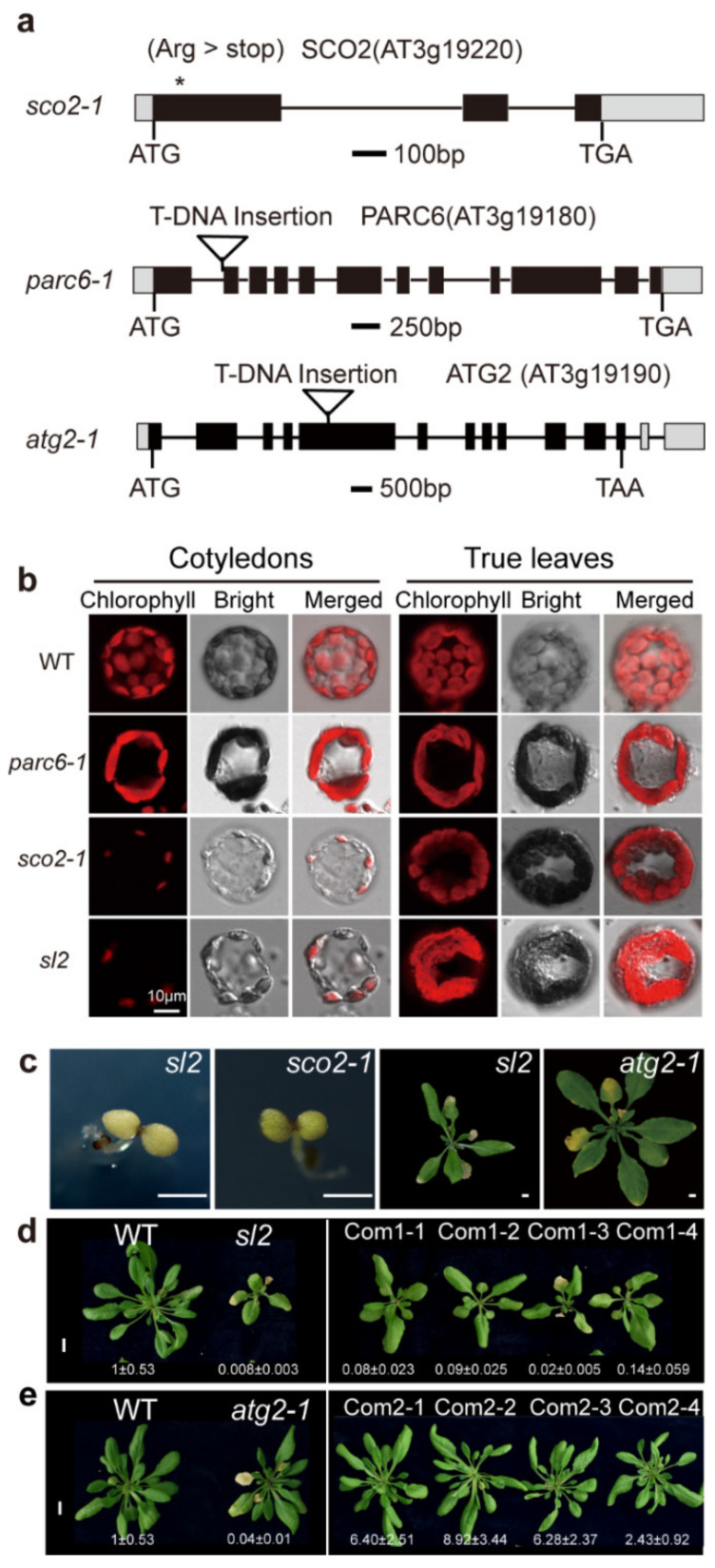
Identification of the corresponding genes for the mutant phenotypes. (**a**) Genomic structure models of *SCO2*, *PARC6* and *ATG2*. Black rectangles show the open reading frame, black lines show introns, and gray rectangles show untranslated regions (UTR). Asterisks mark pointed mutation; triangles represent T-DNA insertion; (**b**) Confocal microscopy observations of chloroplasts in six-day-old cotyledons and 14-day-old true leaves; (**c**) Seven-day-old *sl2* and *sco2-1* mutants grown on ½ MS medium supplemented with 1% sucrose (**left**); five-week-old *sl2* and *atg2-1* mutants grown in soil (**right**). Bar = 25 mm; (**d**) The *sl2* complementary lines expressing *ATG2* cDNA driven by 35S promoter grown in soil for five weeks. The relative expression levels of *ATG2* are shown under the plants. Bar = 1 cm; (**e**) The *atg2-1* complementary lines expressing *ATG2* cDNA driven by 35S promoter grown in soil for five weeks. The relative expression levels of *ATG2* are shown under the plants. Bar = 1 cm.

**Figure 4 ijms-21-04515-f004:**
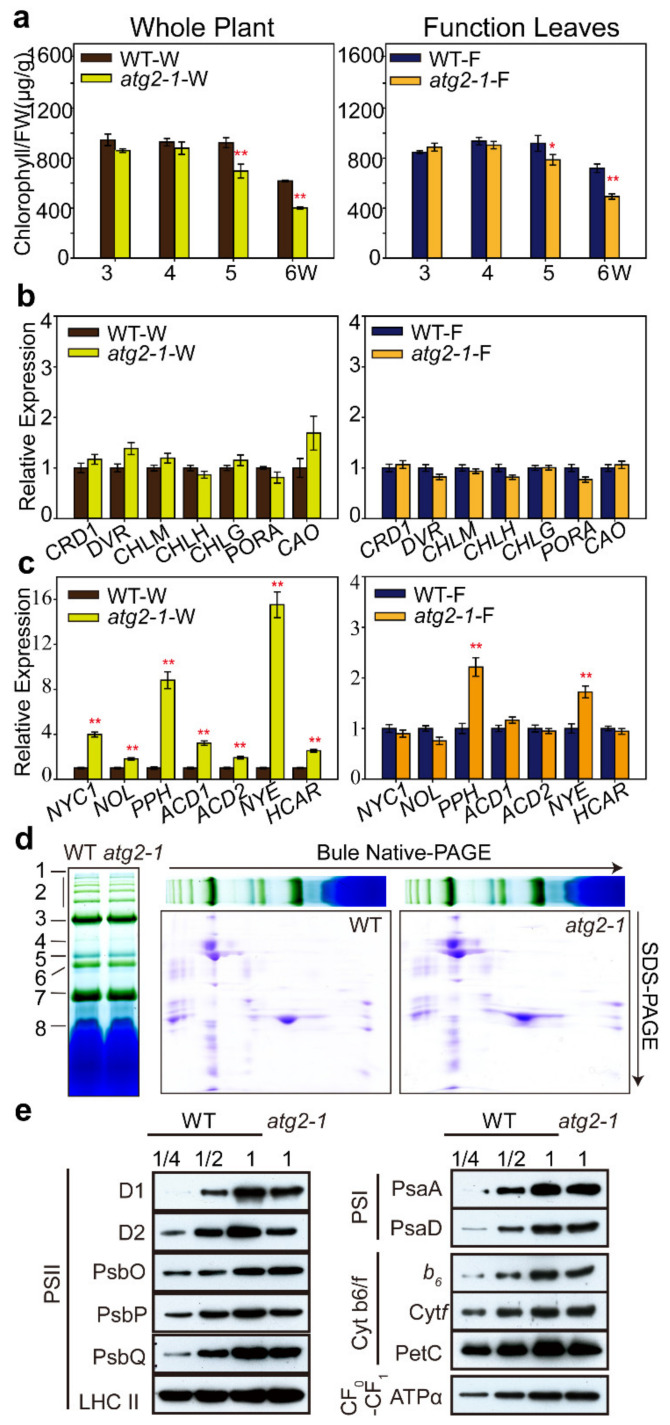
Lack of *ATG2* accelerated degradation of chlorophyll and thylakoid proteins. (**a**) The chlorophyll contents of whole plants (left) and functional leaves (right) of WT and *atg2-1* mutant plants grown for three to six weeks in soil. Values are mean ± SE (*n* = 6 plants). Asterisks show significant difference from the WT plants at * *p* < 0.05 and ** *p* < 0.01 (Student’s *t* test); (**b**) Expression levels of chlorophyll synthesis-related marker genes measured by quantitative RT-PCR in whole plants and functional leaves; (**c**) Expression levels of chlorophyll degradation-related marker genes measured by quantitative RT-PCR in whole plants and functional leaves. The results are shown as mean ± SE from three experiments. Asterisks show significant difference from the WT plants at ** *p* < 0.01 (Student’s *t* test); (**d**) Blue native polyacrylamide gel and the subsequent second dimension SDS-PAGE analysis of thylakoid proteins from five-week-old WT and *atg2-1* mutant plants (10 wild-type plants and 10 *atg2-1* plants were used in this experiment). Annotation of the different complexes: 1, NDH-PSI; 2, PSII super complexes; 3, PSI monomer, PSII dimer and PSII monomer with LHCII trimers; 4, PSI monomer and CF_1_ complex; 5, PSII monomer; 6, LHCII assembly; 7, LHCII trimers; 8, LHCII monomers; (**e**) Analysis of thylakoid membrane proteins accumulation in the five-week-old WT and *atg2-1* mutant plants.

**Figure 5 ijms-21-04515-f005:**
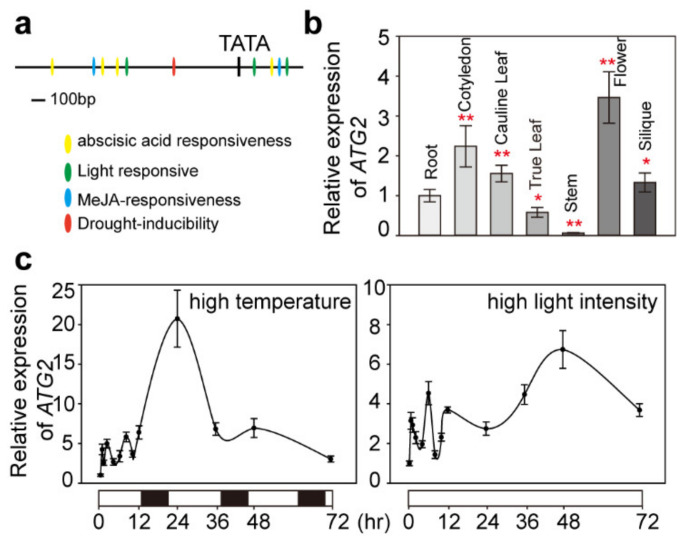
Cis-acting regulatory and expression pattern of *ATG2*. (**a**) Cis-acting regulatory elements analyzed by Plant CARE. The bar represents 100 bp of nucleic acids; (**b**) Expression pattern of *ATG2* in various organs analyzed via quantitative RT-PCR. Values are mean ± SE. Asterisk indicates significant difference from the wild type at ** *p* < 0.01 (Student’s *t* test); (**c**) Expression levels of *ATG2* induced by high light intensity and high temperature stresses. WT plants were grown on ½ MS medium containing 1% sucrose under normal conditions for 10 days, then exposed to high light intensity (350 μmol·m^−2^·s^−1^) or high temperature (30 °C). White bars denote lighted intervals and black bars denote darkness.

**Figure 6 ijms-21-04515-f006:**
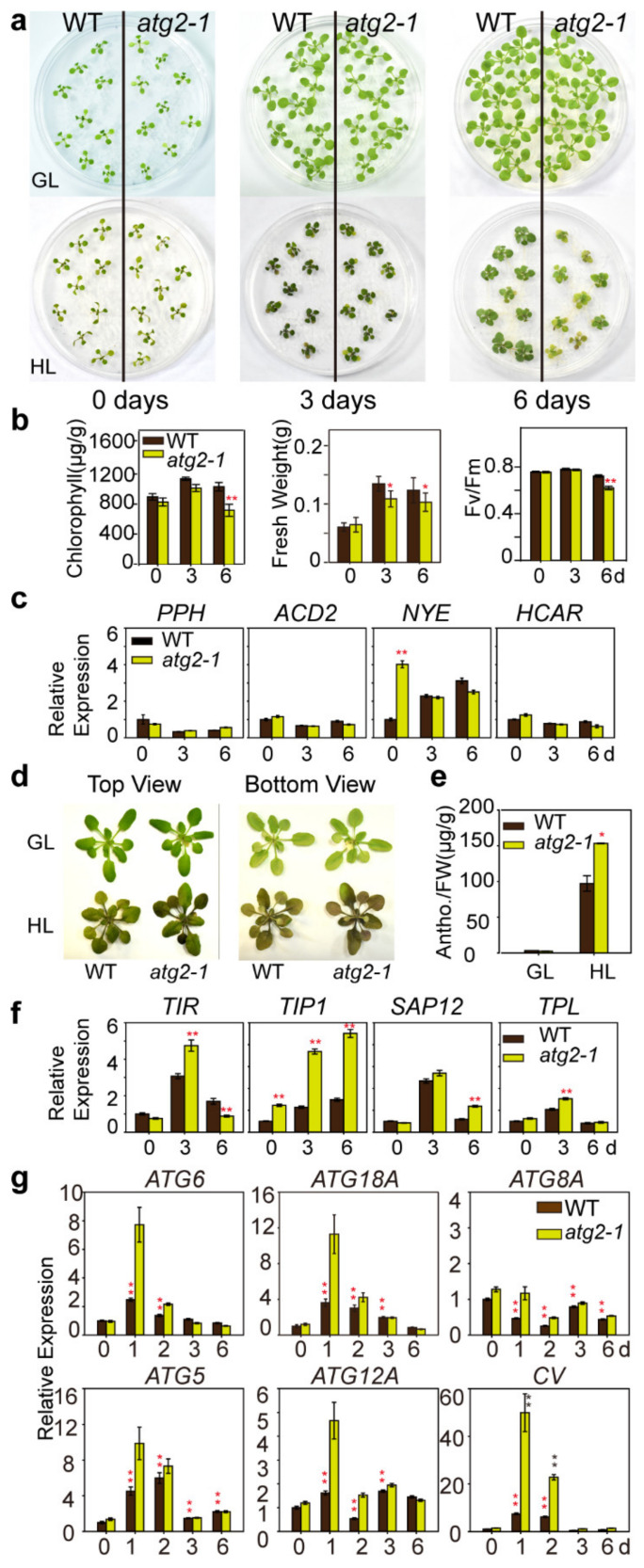
Characterization of *ATG2*-deficient plants under high light intensity conditions. (**a**) Ten-day-old WT and *atg2-1* mutant plants grown under growth light intensity (GL, 80 μmol·m^−2^·s^−1^) and high light intensity (HL, 350 μmol·m^−2^·s^−1^) conditions for three and six days; (**b**) The chlorophyll contents, fresh weight and chlorophyll fluorescence of WT and *atg2-1* plants treated by high light intensity. Values are mean ± SE (*n* = 4 for chlorophyll contents, *n* = 8 for fresh weight, and *n* = 8 for chlorophyll fluorescence). Asterisks show significant difference from the wild type at ** *p* < 0.01 (Student’s *t* test); (**c**) Expression levels of chlorophyll degradation marker-related genes were measured by quantitative RT-PCR before and after high light treatment. Asterisks show significant difference from the wild type at * *p* < 0.05 and ** *p* < 0.01 (Student’s *t* test); (**d**) Three-week-old WT and *atg2-1* plants grown under growth light (GL) or high light intensity (HL) conditions for two days; (**e**) The anthocyanin content of WT and *atg2-1* in (**d**). Values are mean ± SE (*n* = 4 plants). Asterisk indicates significant difference from the wild type at * *p* < 0.05 (Student’s *t* test); (**f**) Expression levels of reactive oxygen species (ROS) marker genes measured by quantitative RT-PCR before and after high light intensity treatments. Asterisk indicates significant difference from the wild type at ** *p* < 0.01 (Student’s *t* test); (**g**) Expression levels of *ATG* genes and *chloroplast vesiculation* (*CV*) in WT and *atg2-1* seedlings measured by quantitative RT-PCR before and after high light intensity treatment. Red asterisk indicates significant difference from the 0-day wild type at ** *p* < 0.01 (Student’s *t* test). Black asterisk indicates significant difference from the wild type at ** *p* < 0.01 (Student’s *t* test).

**Figure 7 ijms-21-04515-f007:**
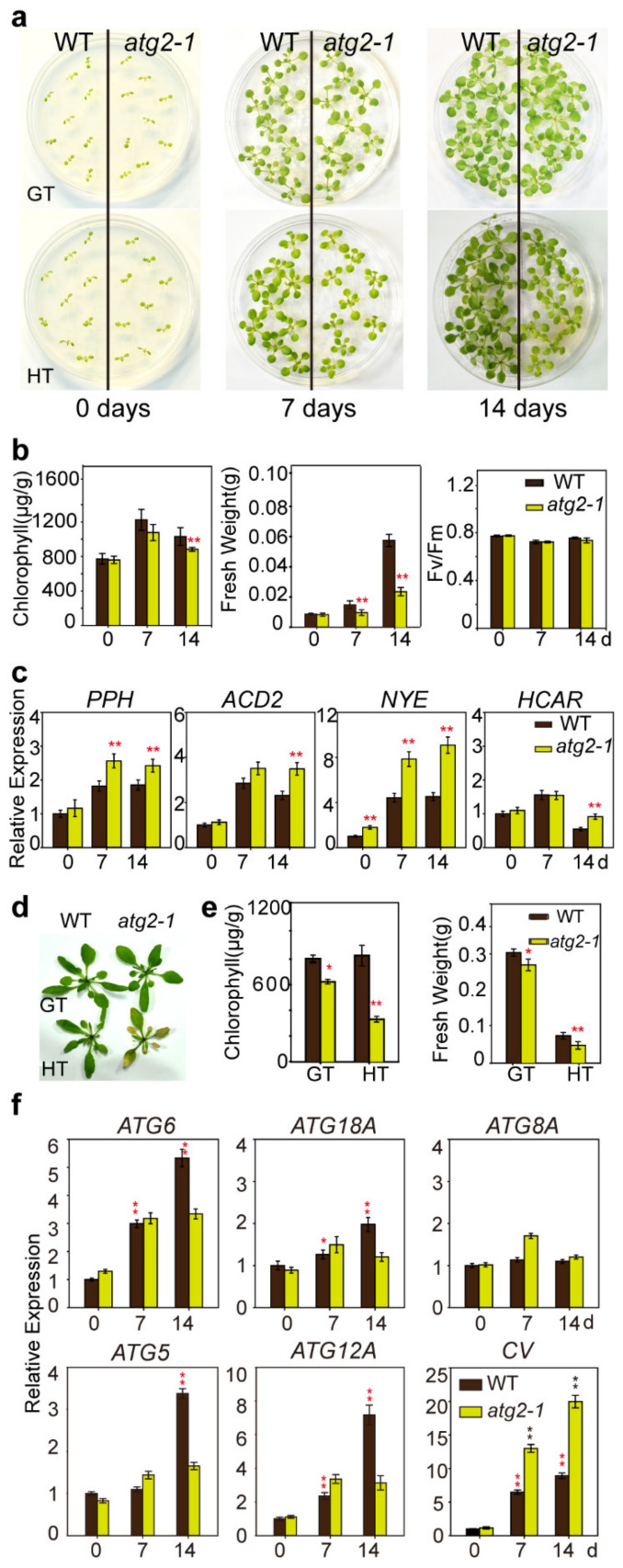
Characterization of *ATG2*-deficient plants under high temperature conditions. (**a**) Seven-day-old WT and *atg2-1* mutant plants grown under high temperature (HT, 30 °C) and growth temperature (GT, 22 °C) conditions for seven and 14 days; (**b**) The amounts of chlorophyll, fresh weight, and chlorophyll fluorescence of WT and *atg2-1* plants treated by high temperature. Values are mean ± SE (*n* = 4 for chlorophyll contents, *n* = 8 for fresh weight, and *n* = 8 for chlorophyll fluorescence). Asterisks show significant difference from the wild type at ** *p* < 0.01 (Student’s *t* test); (**c**) Expression levels of chlorophyll degradation-related marker genes measured by quantitative RT-PCR before and after high temperature treatment. Asterisks show significant difference from the wild type at ** *p* < 0.01 (Student’s *t* test); (**d**) Three-week-old WT and *atg2-1* plants grown under high temperature (HT, 30 °C) and growth temperature (GT, 22 °C) conditions for seven days; (**e**) The chlorophyll content and the fresh weight of WT and *atg2-1* in (**d**). Data are shown as mean ± SE (*n* = 4 for chlorophyll contents and *n* = 6 for fresh weight). Asterisk indicates significant difference from the wild type at * *p* < 0.05 and ** *p* < 0.01 (Student’s *t* test); (**f**) Expression levels of *ATG* genes and *chloroplast vesiculation* (*CV*) in WT and *atg2-1* seedlings measured by quantitative RT-PCR before and after high temperature treatment. Red asterisk indicates significant difference from the 0-day wild type at ** *p* < 0.01 (Student’s *t* test). Black asterisk indicates significant difference from the wild type at ** *p* < 0.01 (Student’s *t* test).

**Figure 8 ijms-21-04515-f008:**
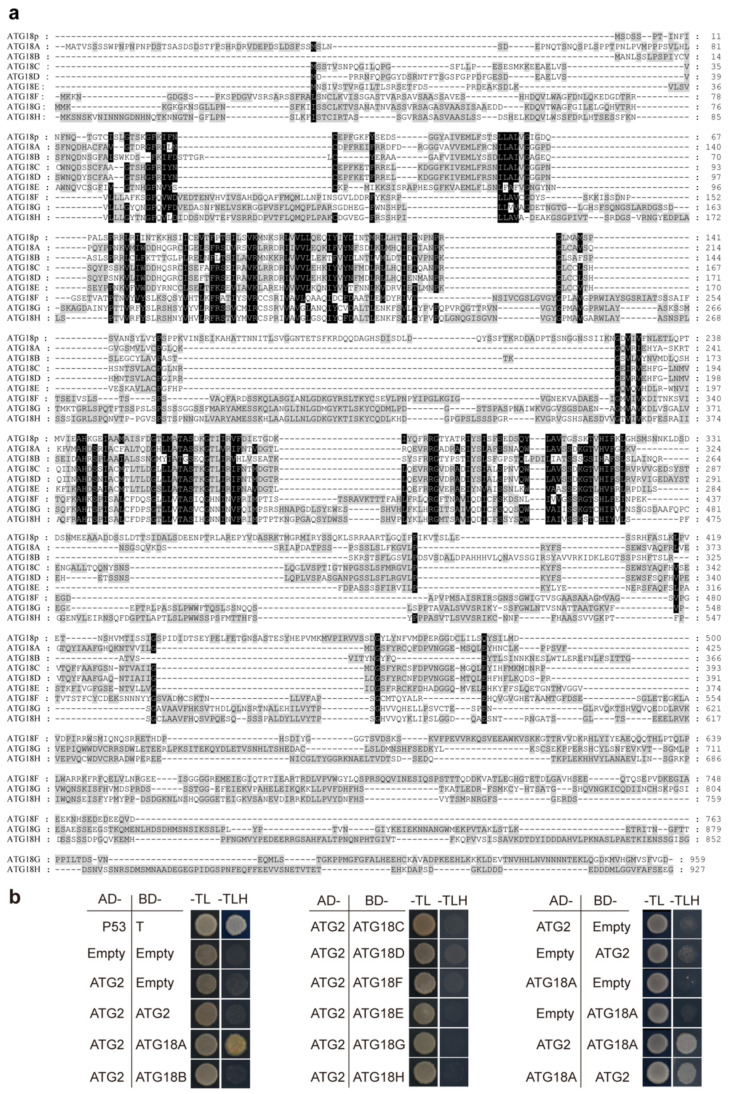
AtATG2 interacts with AtATG18A but no other family members of ATG18 in a yeast two-hybrid system. (**a**) Alignment of protein sequence of ATG18p in *Saccharomyces cerevisiae* and ATG18A-H in *Arabidopsis*. Full-length protein sequences were retrieved from NCBI (https://www.ncbi.nlm.nih.gov/) and aligned in following multiple sequence alignment with MEGA5.2; (**b**) Yeast two-hybrid analysis of interactions of AtATG2 with AtATG2 and ATG18 family proteins including ATG18A, ATG18B, ATG18C, ATG18D, ATG18E, ATG18F and ATG18H. Full-length *ATG2* CDS was fused to the AD domain and co-expressed with the indicated ATG-BD proteins in yeast strain YH109. The positive clones were selected on SD medium with 1mM 3-AT lacking Trp, Leu and His (–TLH). AD-E indicates empty AD plasmid, BD-E indicates empty BD plasmid, pGADT7-T + pGBKT7-53 was a positive control.

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
