# Peer review of "Autophagy-Related 2 Regulates Chlorophyll Degradation under Abiotic Stress Conditions in Arabidopsis"

_ijms, 2020, doi:10.3390/ijms21124515_

Round 1
Reviewer 1 Report
The manuscript by Jiang et al. deals with the analysis of mutants in chloroplast development and metabolism. Photosynthesis is the central pathway leading to autotrophy of plants. With respect to a necessary adaptation of plants to a changing environment, increased efforts towards a detailed understanding of chloroplast metabolic processes is needed. Novel mutants are valuable tools in such analysis.
The authors identified a novel mutant, sl2 and could identify a 118 kb deletion, covering 37 genes. Among them known genes encoding proteins involved in chloroplast function (sco2, parc6 and atg2). Here the authors should add gene names to TableS1, when they are known.
The sco2 mutant very much resembles the sl2 mutant. However, the question about the main reason for the sl2 mutant phenotype, meaning which of the deleted genes is mainly responsible in this respect, remains more or less open. The authors should discuss this in more detail. In addition, the reason why sl2 seeds cannot grow without addition of sucrose remains obscure. Again, this should be discussed. Otherwise the introduction of the sl2 mutant and the corresponding work is merely only a tool to explain why the rest of the paper deals with ATG2 and analysis of the atg2-1 mutant. As seen in Fig. 3d, Complementation of sl2 by ATG” overexpression is only partially successful.
Nevertheless the authors switch topics and concentrate on a role of ATG2 in chlorophyll metabolism, arguing that ATG2 is responsible for the observed reduced chlorophyll contents in older sl2 mutant plants (early senesence, also described for ATG2 mutants). To support this I recommend to include the sco2 mutant in the analysis of chlorophyll leaves (for example in Fig. 4).
Points to consider:
- Why are expression levels in the complementation lines (Fig. 3d) so low (lower than WT)?
- I think it is not justified to conclude a compensatory up-regulation of the CV pathway in atg2-1 only on increased CV gene expression, although the concept is plausible.
- Please indicate which gene is analyzed in Fig. 5
- Define what is SSGL and PAS?
- Give the source of the antibodies used
Reviewer 2 Report
In M&M part Authors should add information about number of analyzed plants.
RT PCR assays - „All experiments were performer for each biological replicate” – I think the most important information is number of biological replicate and number of technical replicate. Authors should add this information.
Information from results part that ½ MS medium (half-concentration of normal MS) should be in M&M part.
Add information about genes responsive to ROS (name, function).
In figures legend:
„Results from one of three independent experiments were shown.” Authors performed 3 independent experiments with three biological replicates for each and showed results for one? Or only for one of three biological replicate showed the resultS? I think this information should be more precise.
(e) The expression analysis of the genes indicated in (c)by RT-PCR.A, actin2. In this legend Authors should indicates genes - number, not indicate in (C). For me is unclear. This is also for many others figures legend.
Fig. 1 b. The left part of panel (table) is a description for pictures on the right side? Maybe Authors are able to change this? I think it is not clear and is hard to read.
Fig 2e. 3d The numbers should be smaller, in this version is very hard to read.
Fig. 5b. Statistical analysis?
Table S1. I think that Authors should add the numbers.
The discussion should be improved. In my opinion Authors should be more focused on results and not on presentation of literature data.
Reviewer 3 Report
see attached

Reviewer 4 Report
This work identified an Arabidopsis mutant, sl2, which shows defective chloroplast and accelerated chlorophyll degradation and revealed that loss-of-function mutation of ATG2 autophagy gene is responsible for the phenotype. The claim of this manuscript is very significant, but I raise some concerns to be corrected or addressed before publication as below.
Major points
“Materials and and Methods” section is surprisingly not enough.
Line 192. Antibodies were generated and used for detecting some proteins. However, there is no description of their detail such as antigens and origin (mouse, rabbit or others?). Authors should provide clear information of them.
Line 215. “a collection of mutant lines” authors should reveal how authors made the mutant lines and what type of mutations they have in M&M section.
Lines 165 and 272. “Bulked segregant analysis” There is no description about its detail. Authors should provide the detailed BSA methodology in M&M section.
Statistics is required to confirm significant differences. However, some figures showing gene expression such as Figs. 4c lacked statistics. Please add statistics.
Minor points
Line 24. “a novel Arabidopsis mutant, sl2” is better.
Line 28. “ATG2” should be italic.
Line 184. Please indicate how many replicates of qRT-PCR were performed.
Line 189. “chloroplast protein extraction buffer” Please clarify the composition.
Line 239. “The true leaf … enlarged chloroplasts” It is hard to confirm this description only by EM image (Fig. 1e). Is it possible to provide data of chloroplast size?
Lines 248-250. The sentence “The chlorophyll was … proteins were affected” may be prediction. Please rephrase it with more appropriate one.
Line 278. Fig. 2e. Why were lengths of PCR products same among genes? Were PCR cycle numbers also same?
Line 406. “CV-related gene” What is it? I know it is AT2G25625. It is better to introduce the gene by citing an appropriate reference.
Line 391. In addition, it may be better to show gene names but not IDs for genes in Fig. 6f.
Line 450. “ATG18A mutants showed …” may be correct.
Reviewer 5 Report
Comments and Suggestions for Authors
The paper entitled “Autophagy-related 2 regulates chlorophyll degradation under abiotic stresses conditions in Arabidopsis”. The authors identified and characterized a novel mutant sl2, showing defective chloroplast development and accelerated chlorophyll degradation.
This work is interesting but unfortunately, the manuscript is not properly prepared for the Journal. Probably the authors have not read the “instruction for the authors”. The entire manuscript should be corrected.
It is mandatory to correct the manuscript in some points:
# line 24: define “sl2”
# line 31: I suggest remove the short name “CV” and use normal sentence in all article - “chloroplast vesiculation”.
# keywords: I suggest remove the dots and put dashes.
# All literature in the text to be improved. Read the instruction for the authors“.
# line 158: between “0.2” and “g” insert space.
# line 165: I suggest change the sentence and shoul be: “…The large fragment deletion locus was identified by map base cloning combined with Bulk Segregant Analysis (BSA).”.
# line 187: “…was performed as described ….” – by who? Correct it.
# line 202: should be “…medium containing 50 mg/mL hygromycin B.”.
# line 352: should be “... and high temperature (30 °C).”.
# line 416: should be “…different from WT plants under the normal temperature (22 °C).”
# line 426: between 30 and °C insert space.
# references: should be improved. Read the instruction for the authors“.
Round 2
Reviewer 1 Report
Revision of the manuscript led to a improvement of the work which is now suitable for publication
Reviewer 5 Report
Now the article is ready to publish.